# Selenium Supplementation and Prostate Health in a New Zealand Cohort

**DOI:** 10.3390/nu12010002

**Published:** 2019-12-18

**Authors:** Nishi Karunasinghe, Lance Ng, Alice Wang, Venkatesh Vaidyanathan, Shuotun Zhu, Lynnette R. Ferguson

**Affiliations:** Auckland Cancer Society Research Centre (ACSRC), Faculty of Medical and Health Sciences (FM&HS), The University of Auckland, 85 Park Road, Auckland 1023, New Zealand; lance.rab.ng@gmail.com (L.N.); alice.wang@auckland.ac.nz (A.W.); v.vaidyanathan@auckland.ac.nz (V.V.); stzhu@hotmail.com (S.Z.); l.ferguson@auckland.ac.nz (L.R.F.)

**Keywords:** selenium supplementation, prostate-specific antigen (PSA), genotypes, dietary nutrients

## Abstract

Background: There is variable reporting on the benefits of a 200 μg/d selenium supplementation towards reducing prostate cancer impacts. The current analysis is to understand whether stratified groups receive supplementation benefits on prostate health. Methods: 572 men were supplemented with 200 µg/d selenium as selinized yeast for six months, and 481 completed the protocol. Selenium and prostate-specific antigen (PSA) levels were measured in serum at pre- and post-supplementation. Changes in selenium and PSA levels subsequent to supplementation were assessed with and without demographic, lifestyle, genetic and dietary stratifications. Results: The post-supplementation selenium (*p* = 0.002) and the gain in selenium (*p* < 0.0001) by supplementation were significantly dependent on the baseline selenium level. Overall, there was no significant correlation between changes in PSA and changes in selenium levels by supplementation. However, stratified analyses showed a significant inverse correlation between changes in PSA and changes in selenium in men below the median age (*p* = 0.048), never-smokers (*p* = 0.031), men carrying the GPX1 rs1050450 T allele (CT, *p* = 0.022 and TT, *p* = 0.011), dietary intakes above the recommended daily intake (RDI) for zinc (*p* < 0.05), and below the RDI for vitamin B12 (*p* < 0.001). Conclusions: The current analysis shows the influence of life factors on prostate health benefits of supplemental selenium.

## 1. Introduction

There is an increasing incidence in prostate cancer worldwide, thought to be due to multiple reasons, including that of a western dietary lifestyle [1,2,3]. Studies have been attempted to provide dietary solutions for prostate cancer as reviewed by DiMarco-Crook et al., 2015, including the joint application of selenium with other nutrients [4]. Meanwhile, there is variable reporting on the benefits of a 200 μg/d selenium supplementation towards reducing prostate cancer risk [5,6]. A reduction in cancer incidence including that of prostate cancer by selenium supplementation was shown with 200 µg/d selenized yeast as the supplementation mode over a mean supplementation period of 6.4 years in a cohort of men with a 115 ng/mL baseline median serum Se level in the Nutrition Prevention of Cancer (NPC) study [5]. An extended NPC study analysis with a mean supplementation period of 7.4 years showed similar but attenuating benefit trends with overall and prostate cancer incidence, while showing an increased risk of some other cancers, [6] and increased diabetes risk among those with baseline selenium levels above 121.6 ng/mL [7]. Meanwhile, no benefits were observed by way of prostate cancer reduction when 200 µg/d selenium as selenomethionine was supplemented for 5.5 years in a cohort of men with a median baseline serum selenium level of 135 ng/mL in the Selenium and Vitamin E Cancer Prevention Trial (SELECT) [8]. Subsequent analyses from this study have recorded increased overall and high-grade prostate cancer hazards ratios in men within the top two quintiles of baseline toe nail selenium levels, when receiving selenium with 400 IU/day of all-racemic alpha-tocopheryl acetate (AT) as well as in the pooled group of men receiving both selenium and AT and selenium alone [9]. A twelve-year prospective monitoring study of selenium intakes in men initially diagnosed with non-metastatic prostate cancer has shown increased prostate cancer-specific mortality in those consuming over 140 µg/d in the form of selenium containing supplements [10]. The most recent Cochrane systematic review on selenium for preventing cancer concludes that the risk ratio for prostate cancer incidence or mortality has no strong negative association with selenium exposure particularly among randomized control studies [11]. However, the authors view on the SELECT study is that sub-group analyses with specific characteristics including baseline selenium exposure and genetic factors would provide further clarification on selenium impacts on cancer prevention. One such SELECT sub-group study is reported by Martinez et al., 2014 [12], where an association of the NK3 Homeobox 1 (NKX3.1) rs11781886 CC genotype, combined with selenium administration, is shown to increase both overall prostate cancer risk as well as low grade prostate cancer risk. These authors further report that men carrying both the NKX3.1 rs11781886 CC and CT genotypes have increased prostate cancer risk when supplemented with vitamin E.

In New Zealand, prostate cancer patients record lower serum selenium levels (101.2 ± 1.01 ng/mL) at diagnosis compared to New Zealand men without prostate cancer diagnoses (112.9 ± 1.01 ng/mL) [13], and prostate cancer patients from the USA (135 ± 21 ng/mL and 132 ± 25 ng/mL for European Americans (EA) and African Americans (AA) respectively) [14]. The levels recorded for New Zealand men with no known diagnoses of prostate cancers [13] is also lower than the levels recorded for similar men from the USA (140 ± 28 and 134 ± 21 ng/mL for EA and AA respectively) [14]. Our studies have also shown that optimal levels of serum selenium towards minimizing DNA damage are recorded between 116 and 150 ng/mL which also have genetic influences [15]. The influence of age and BMI in influencing serum selenium retention was reported by us previously [16]. We have also shown that a 200 μg/d selenium supplementation benefits by way of DNA damage reduction is achieved dependent on dietary methionine and folate levels and BMI [16,17]. Studies including ours have reported selenium supplementation to lower serum prostate-specific antigen (PSA) levels among men with high-risk for prostate cancer and among middle-aged US men [18,19]. Our studies have previously identified several single nucleotide polymorphism (SNP) genotypes including those of selenoproteins that increase the risk for prostate cancer incidence in New Zealand [13,20].

The *glutathione peroxidase 1* (*GPX1*) is a selenoprotein encoding gene located in the chromosomal region 3p21.3 and records rs1050450 C>T SNP that is responsible for a nonsynonymous amino acid change from proline to leucine [21,22]. The catalytic activity of GPX is affected by the rs1050450 T allele [23], while this allele is also associated with several cancers [13,24,25]. The selenoprotein 15 (SEL15) encoding gene is located in chromosomal region 1p31 and records rs5845 C>T SNP [26,27]. This rs5845 C>T SNP which relates to the amino acid position 811 is located in a selenocysteine insertion sequence (SECIS)-like structure within the SEL15 protein, and is associated with the rs5859 G>A SNP which relates to the amino acid position 1125 located within the SECIS element [27]. According to these authors, the latter polymorphism influences selenocysteine incorporation in SEL15. The SEL15 rs5845 and its associated SNP rs5859 are associated with several cancers [13,28,29]. The mitochondrial manganese superoxide dismutase (MnSOD) encoding gene is located in the chromosomal region 6q25.3 and the rs4880 C>T SNP produces a nonsynonymous alanine to valine amino acid change [30,31]. This rs4880 polymorphism also has been associated with various cancers [32,33]. The aldo-keto reductase 1C3 (AKR1C3) encoding gene is located in the chromosomal region 10p15 and the rs12529 C>G polymorphism produces a nonsynonymous amino acid change from histidine to glutamine [34]. This rs12529 polymorphism is also associated with various cancers [20,35,36]. The *kallikrein-related peptidase 3* (*KLK3*) gene encodes the serine protease PSA and this gene is located in the chromosomal region 19q13 [37]. The *KLK3* rs17632542 T>C is a nonsynonymous polymorphism with an amino acid change from isoleucine to threonine and has an association with prostate cancer risk as well as with the PSA level [20,38].

Here, we present an analysis that looked into variable utilization of 200 μg/d selenium in the form of a selenized yeast supplement for six months, and subsequent benefit variation in maintaining prostate glandular epithelium microarchitecture stability, as measured through the surrogate marker serum PSA. This is a stratified analyses based on demographic and lifestyle characteristics, baseline serum selenium levels and a set of genotypes known to associate with prostate cancer risk [13,20,39,40], as well as a panel of dietary nutrient factors associated with DNA methylation [41,42,43].

## 2. Materials and Methods

### 2.1. Participant Recruitment, Blood Sample Collection and Selenium Supplementation

At the design and ethics application stages of this supplementation study in 2005, there was no requirement for registering this study with a clinical trials registry. The Australian New Zealand Clinical Trial Registry (ANZCTR) reached its recognition from the World Health Organisation in 2007 [44]. Therefore, this selenium supplementation study is retrospectively registered with the ANZCTR. The related trend statement with trend checklist is provided as Appendix A.

A cohort of men (*n* = 572), self-reported as having no history of cancers (other than non-melanoma skin cancers), and not taking more than 50 µg selenium/day as supplements were recruited to a selenium supplementation study [15,16,45]. Participant recruitment was carried out with informed consent (ethics reference NTY/06/07/060, from the Health and Disability Ethics Committees, Ministry of Health, New Zealand). Recruitment of men to this study started on the 16 October 2006 and completed on the 22 December 2008. The last post-supplementation study visit was on the 13 August 2009. At the baseline study visit, the height and weight of each participant was measured and recorded at the study centre based at the Faculty of Medical and Health Sciences, University of Auckland, New Zealand.

These men were of the age range ≥20 years to <80 years. Participants provided baseline blood samples in a Becton Dickinson (BD, Franklin Lakes, New Jersey, United States) plain vacutainer tube for subsequent serum collection and a BD ethylene diamine tetra acetic acid (EDTA) vacutainer tube for subsequent DNA extraction. A health and lifestyle questionnaire was administered at study entry to collect participant information including age, tobacco smoking and alcohol consumption lifestyle and health status including urology issues. Serum samples were separated and collected within two hours of blood draw by centrifugation at 2000× *g* for 10 min at 4 °C in an Eppendorf 5810R centrifuge from Hamburg, Germany. Serum aliquots were stored in −80 °C until serum selenium and PSA levels were measured.

### 2.2. Serum Selenium Measurements

The serum levels of selenium were measured in batches in all baseline samples within two months of sample collection. These measurements were made at the Gribbles Veterinary Pathology, Hamilton, New Zealand using a modified semi-automated fluorometric assay based on Watkinson [46], Watkinson and Brown [47] and Rongpu et al. [48]. The fluorescence of the final benzopiazselenol extracted into cyclohexane was measured with an excitation wavelength of 360 nm and emission wavelength of 518 nm. The selenium assay recorded 2.6% intra assay coefficient of variation and 11.2% inter assay coefficient of variation.

### 2.3. Selenium Supplementation

Men were stratified based on their baseline serum selenium levels ≥200 ng/mL or <200 ng/mL. Only those with serum levels of <200 ng/mL were supplemented with 200 µg/d selenized yeast from Alltech Dunboyne, Ireland, (sourced from Republic of Serbia) for a period of six months. At the post-supplemented time point, they provided another blood sample in a BD plain tube for post-supplemented serum collection procedure as mentioned before. The selenium levels of these post-supplementation serum samples were measured as before.

### 2.4. SNP Genotyping

Total genomic DNA was extracted from EDTA blood collected within 24 h, using the QIAamp genomic DNA kit (Qiagen, Hilden, Germany) following the manufacturers’ protocol with the aid of a fully automated QIAcube (Qiagen, Hilden, Germany). Extracted DNA was quantitated using a Nanodrop 1000 v.3.8. (Thermo Scientific, Wilmington, USA), and stored at −20 °C until used for genotyping. Genotyping was carried out using the Sequenom or Taqman methods as described previously [15,49]. The panel of genetic polymorphisms selected for this analysis were—*GPX1* rs1050450, *SEL15* rs5845, *AKR1C3* rs12529 and *KLK3* rs17632542. based on our own studies from New Zealand representing genetic risk association for prostate cancer [13,20]. The *MnSOD* rs4880 SNP was selected based on its impacts on lethal prostate cancer [40] as well as advanced prostate cancer when interacting with toe nail selenium [39].

### 2.5. Serum PSA Measurements

In December 2012, total PSA was measured from remaining baseline and post-supplemented serum aliquots stored at −80 °C. These measurements were carried out at the LabPlus, Auckland, New Zealand using electrochemiluminescence immunoassay (Roche Cat. #. 04641655 190) on a Roche Modular E170 anaylser (Roche Diagnostics, NZ). Total assay imprecision was 3.2% at a level of 1.12 ng/mL, 3.7% at 4.61 ng/mL, and 2.7% at 27.5 ng/mL.

### 2.6. Diet and Activity Data Analysis

At study entry, participants were provided with four-day diet and activity diary forms with specific instructions on completing them on four consecutive days, preferably including one weekend day. Participants were requested to select their regular activity pattern, from seven categories given with the dietary recording sheets. These included, very sedentary, sedentary, light, light moderate, moderate, heavy and very heavy. The four-day diet and activity diary data along with the BMI estimated at baseline and age at recruitment were uploaded to the FoodWorks Professional Version 9, (Xyris Software, Queensland, Australia, Pty, Ltd). It was noted that some items listed in the food diaries lacked details in terms of the food item itself and the quantity. These data were independently checked for possible variable entries of food and drink quantity and quality. Based on these checks, a list of such commonly variable-entries on items and unknown quantities transferred to the FoodWorks Database were identified and listed. Default entries to be used in such circumstances were listed (Appendix A). General rules for FoodWorks data entry were also written up (Appendix A). Using the above table and rules, all data entries without sufficient information on quantity and quality of dietary items were independently standardized. Records that lacked sufficient information or provided unacceptable recordings (e.g., basal metabolic rate being higher than the energy provided by the diet) were removed from the analysis. 192 participants failed to provide diet and activity data recordings while 37 were removed from analyses due to lack of adequate information. Therefore, only 343 diet and activity records were considered for subsequent analyses.

### 2.7. Statistical Analysis

Continuous variables of age, BMI, serum selenium and PSA levels between all participants at baseline, and those completing the supplementation protocol, were tested using the Mann–Whitney Rank Sum Test, as the data were not normally distributed. Categorical variables of tobacco smoking and alcohol consumption lifestyles between the baseline and protocol completing groups were tested with the F-statistics. Distribution of health disorders and the food and activity record submission details of these two groups were tested with the Chi Square statistics. The distribution of genotypes in the tested panel of SNPs were tested for biallelic distribution using the Hardy Weinberg equilibrium calculator [50]. Variation of baseline and post-supplemented levels of serum selenium and serum PSA between genotype groups of each genetic polymorphism were compared using the Analysis of Variance (ANOVA) on Ranks, as the variables were not normally distributed. All correlations were tested using the linear regression with the function f = y0 (intercept) + a (regression coefficient) × X. These correlations were assessed both with and without stratification for demographic, lifestyle, baseline serum selenium, baseline serum PSA, SNP genotypes and dietary nutrient levels. Dietary intakes were stratified based on below or above the recommended dietary intake (RDI) [51] of selenium, zinc, vitamin B12, and dietary folate equivalents, and below or above the median of the % energy derived from dietary proteins. Dietary intake cut-offs for selenium, zinc, vitamin B12 and dietary folate equivalents were therefore considered as 70 µg/d, 14 mg/d, 2.4 µg/d and 400 µg/d respectively as given by the joint report from the National Health and Medical Research Council, Australia and New Zealand Ministry of Health [51]. The cut-off for the energy intake from protein was considered as the median value of 16.0% recorded in the current cohort. The Mann-Whitney Rank Sum Test, ANOVA on Ranks, Chi Square test, F statistics and linear regression test were performed using SigmaPlot version 14.0 (Systat Software Inc., California, USA). A significance level of *p* < 0.05 was set out for all analyses.

The study summary flow chart is presented in Figure 1.

## 3. Results

### 3.1. Summary of Study Participation

Out of 572 men taken part at the baseline evaluations, 568 (99.3%) were given selenium supplements and 481 (85%) completed the six-month supplementation protocol (Figure 1). There were no significant differences between age, BMI, baseline serum selenium and PSA levels, tobacco smoking and alcohol consumption lifestyles, health disorder status and the acceptable food and activity diary submission between those completing the supplementation protocol and all taking part at the baseline evaluation (Table 1).

### 3.2. SNP Genotype Distribution of the Cohort 

The genotype and allele frequencies of the SNP panel used in the study are presented in Table 2. All presented SNP data were within the Hardy–Weinberg (HW) equilibrium.

### 3.3. Influence of Baseline Selenium on Effects of Supplemented Selenium

Overall, there was a significant correlation between the baseline serum selenium level and the post-supplementation serum selenium level Y (post-supplementation serum selenium) = 142.140 – 0.177 × X (baseline serum selenium), with a coefficient of determination (*r*^2^ = 0.020, and *p* = 0.002) (Figure 2). The highest difference in serum selenium due to supplementation recorded in this cohort was 126 ng/mL. Data suggests that the difference between post-supplementation and baseline serum selenium levels also depend on the baseline serum selenium level Y (difference in post-supplementation serum selenium) = 142.554 – 0.828 × X (baseline serum selenium), *r*^2^ = 0.314 at a significance level of *p* < 0.0001 (Figure 3).

#### Influence of Baseline Selenium on Supplemented Selenium Stratified by SNP Genotypes

The baseline and post-supplementation serum selenium correlations stratified by genotypes, are presented in Table 3. The *GPX1* rs1050450 TT, *AKR1C3* rs12529 CG and *SEL15* rs5845 CT showed the strongest significant correlations (*r*^2^ = 0.137, slope = 0.516 and *p* = 0.031; *r*^2^ = 0.080, slope = 0.356 and *p* < 0.0001; and *r*^2^ = 0.030, slope = 0.228 and *p* = 0.026 respectively).

### 3.4. Pre and Post-Supplementation Selenium and PSA Levels Stratified by Genotype

A summary of the baseline and post-supplementation serum selenium and serum PSA levels stratified by genotypes is presented in Table 4. Except for the *SEL15* rs5845 and *KLK3* rs17632542, there were no significant genetic difference in baseline and post-supplementation serum selenium and PSA levels. The homozygous recessive TT genotype of the *SEL15* rs5845 genotype recorded the highest post-supplementation serum selenium level of 180.8 ng/mL compared to 157.9 ng/mL and 165.4 ng/mL recorded for the CC and CT genotypes respectively (*p* = 0.03). The baseline serum PSA level of the *KLK3* rs17632542 TT genotype was 0.9 ng/mL compared to 0.6 ng/mL recorded by CT genotype (*p* < 0.01). Due to the insufficient numbers of men with the *KLK3* rs17632542 CC genotype, they were not included in this assessment. The SEL15 rs5845 TT genotype showed a trend *p* = 0.06) towards higher serum PSA at baseline (1.8 ng/mL) compared to CC (0.9 ng/mL) and CT (0.95 ng/mL) genotypes.

### 3.5. Change in Selenium vs. Change in PSA

#### 3.5.1. Overall Change in Selenium vs. Change in PSA

The linear regression between the difference in the serum selenium (X) by supplementation and subsequent change in serum PSA (Y) in the overall group showed no significant correlation (Y = 0.0796 − 0.0044 × X, *r*^2^ = 0.002, and *p* = 0.369).

#### 3.5.2. Stratified Analyses of Change in Selenium vs. Change in PSA

##### Stratified Analysis of Change in Selenium vs. Change in PSA by Demographic, Lifestyle and Baseline Characters

The linear regression statistics between the difference in serum selenium and subsequent change in PSA level due to supplementation stratified by baseline age, BMI, serum selenium, PSA as well as tobacco smoking and alcohol consumption lifestyle are given in Table 5. These correlations were significant only among men below the median age of 55 years (Y = 0.09 − 0.002 **×** X, *r*^2^ = 0.02 and *p* = 0.048) and men who were never-smokers (Y = 0.2103 − 0.003 **×** X, *r*^2^ = 0.02 and *p* = 0.031).

##### Stratified Analysis of Change in Selenium vs. Change in PSA by SNP Genotypes

The linear regression statistics between changes in serum selenium and subsequent changes in serum PSA due to supplementation, when stratified by genotypes are presented in Table 6. Men carrying the variant T alleles of the *GPX1* rs1050450 have a significant inverse correlation between the change in serum PSA and the change in serum selenium due to supplementation. The linear regression equation for the *GPX1* rs1050450 CT genotype was Y (change in serum PSA) = 0.260 − 0.006 × X (change in serum selenium), (*r*^2^ = 0.031, *p* = 0.022). The linear regression equation for the *GPX1* rs1050450 TT genotype was Y = 0.976 - 0.0171 × X, (*r*^2^ = 0.223, *p* = 0.011). The rest of the genetic stratifications showed no significant correlation between the change in serum selenium and the change in PSA level, except for a trend shown with the *KLK3* rs17632542 CT genotype Y = 0.226 − 0.0053 × X, (*r*^2^ = 0.111, *p* = 0.05).

##### Stratified Analysis of Change in Selenium vs. Change in PSA by Dietary Nutrients

The linear regression statistics between the difference in serum selenium levels by supplementation and subsequent changes in serum PSA levels showed an interaction with dietary nutrient levels (Table 7). Especially, men recording dietary zinc intakes above the RDI recorded a significant inverse correlation between the difference in serum selenium levels and subsequent changes in serum PSA levels (*r*^2^ = 0.046, slope = −0.005 and, *p* < 0.05). Dietary intakes of vitamin B12 below the RDI also showed a similar inverse correlation (*r*^2^ = 0.795, slope = −0.020, *p* < 0.001). Although, vitamin B12 levels below the RDI showed the highest significance in PSA reduction after selenium supplementation, these results should be considered with caution due to the lower numbers of men in this category.

Data related to the current analysis is provided as Appendix A.

## 4. Discussion

In an era of highest rate of age standardized global incidence of prostate cancer recorded from New Zealand [52], it is inevitable that we should be looking at every possible source to ease this problem. Restoring general prostate health in New Zealand men could indirectly reduce this long-term public health burden. Meanwhile, New Zealand is known for soil deficiencies in selenium that affects the dietary content of this trace mineral [53]. Our previous studies with New Zealand men have shown that benefits of 200 μg/d selenium supplementation for six months in preserving DNA integrity is achieved only in stratified groups of men [16,45]. We have also shown that DNA integrity benefits vary between serum selenium levels between 116 and 150 ng/mL depending on genetic variation [15]. The current analysis looked into the utilization of supplemented selenium in changing serum selenium levels and subsequent changes in the stability of prostate glandular epithelium microarchitecture, using PSA as a surrogate biomarker.

We observed a significant positive dependence of post-supplementation serum selenium level, on the baseline serum selenium level. When stratified by genotypes, the *GPX1* rs1050450 TT, AKR1C3 rs12529 CG and SEL15 rs5845 CT showed the best correlation between the baseline and post-supplementation serum selenium levels. This implies that the individual serum selenium carrying capacity could be varying with the genotypes.

Beyond the highest gain in serum selenium by supplementation of 126 ng/mL, the overall gain in serum selenium by supplementation declined at a rate of 0.828 ng/mL with each one ng/mL increase in baseline serum selenium level. This means that part of the supplemented selenium would have passed though different routes in addition to replenishing the serum component. Among such possible selenium channeling routes may include, excretion [54]; metabolism by gut biota to forms that reduce absorption into circulation [55]; used up for toxicity modulation [56]; channeled towards other tissue optimization [57] including that of the prostate [58,59,60]; or the selenomethionine component in the selenized yeast [61] getting non-specifically incorporated into a wide range of tissue proteins in place of methionine [62,63]. Takata et al., 2009 have shown that serum selenium level is a reflection of the levels in the prostate [59]. Our previous studies with the current study cohort have shown that selenium supplementation led to no gain in red blood cell *GPX* activity, indicating, this has been optimized with the baseline selenium levels; although the thioredoxin reductases activity was upregulated with selenium supplementation [16,45]. This is in line with the whole body selenium regulation as discussed by Burk and Hill 2015 [64].

There was no overall significant correlation between the difference in serum selenium after supplementation and subsequent difference in PSA level, although significant correlations were recorded among, men below the median age of 55 years as well as the group who were never-smokers. Our previous studies have shown that age based PSA increase in men with prostate cancer beyond 69 years could be due to the increase in extra-testicular androgen production from adrenal dehydroepiandrosterone (DHEA)-based androgen precursors [65] catalyzed by the AKR1C3 enzyme, particularly among men carrying the AKR1C3 rs12529 CG and GG genotypes [66]. Therefore, our current observation of a significant inverse correlation between the difference in serum selenium and difference in PSA level after selenium supplementation among men below the age of 55 years may indicate that increased selenium level does not support a significant reduction in PSA when the extra-testicular androgen production starts to increase with advancing age. A placebo-controlled study tested against 200µg selenium daily for 8 weeks with women aged 10–40 years with polycystic ovarian syndrome has recorded decreased serum DHEA levels with selenium supplementation [67]. This further implies that at an age below 55 years, PSA decline by selenium supplementation may happen due to indirect DHEA suppression. Among ever-tobacco smokers, supplemented selenium may get triaged towards seleno-antioxidant enzyme activity [16] over prostate glandular epithelium stability, restricting prostate health benefits to never-smokers.

When data were stratified by the SNP genotypes, we see a PSA reducing benefit by supplementation only with the *GPX* rs1050450 CT and TT genotypes. The EPIC-Heidelberg nested prostate cancer case-control study indicates that men with the variant T allele of the *GPX1* rs1050450 polymorphism record a risk ratio (odds ratio OR) of 0.87 per 10 ng/mL increase in serum selenium [68]. The authors also showed that the prostate cancer risk ratio was further reduced (OR, 0.64) per 10 ng/mL increase in serum selenium among men with high-grade prostate cancer carrying the rs1050450 variant T allele, while no such risk modifications were observed with increasing serum selenium with the *GPX1* rs1050450 dominant allele C.

PSA is exclusively produced by the glandular prostate epithelium, and is abundant in the seminal fluid [69]. PSA production is signaled by androgen activated androgen receptors [65,67,70]. According to Prcic and co-workers, PSA reaches serum when the microarchitecture of prostate glandular tissue is disrupted [71]. According to these authors increased levels in circulation is an indication of prostate disease or trauma in the prostate gland including prostate cancer. Therefore, the decline in PSA with increase in serum selenium among men carrying the *GPX1* rs1050450 T allele in our study is an indirect reflection of prostate health benefits that may parallel prostate cancer risk reduction as recorded by the Steinbrecher et al. study [68].

In a previous study, an inverse trend in PSA change with increasing baseline serum selenium level was recorded by us for a group of New Zealand men with benign prostate issues but with negative biopsies, and carrying a PSA >4 ng/mL at recruitment [19]. When these men were supplemented with either a 200 or 400 µg/d selenium as selenized yeast or placebo for six months, two thirds of the cohort showed a reduction in the rate of PSA progression. Zhang et al., 2011 reported that a 3-month supplementation with 200 μg/d selenium (in the form of glycinate) increased plasma and erythrocyte glutathione peroxidase (*GPX*) and lowered serum PSA levels [18]. The Procomb study from Italy, where men with lower urinary tract infections (UTI) randomized to receive either a Serenoa repens plant extract containing 50 µg of selenium and 5 mg of lycopene or a placebo for over one year showed no significant change in PSA between the test and the placebo groups [72]. It is a possibility that this combination supplement with 50 µg/d of selenium was not sufficient to influence a pharmacological benefit of selenium in PSA reduction in these men with UTI. A six-month 55 µg/d of selenium supplementation with 35 mg/d of lycopene and 600 mg/d of polyphenol supplementation is reported in a group of men diagnosed with isolated high-grade prostatic intraepithelial neoplasia HGPIN on biopsy [73]. These authors record increased incidence of prostate cancer and markers indicating prostate cancer progression in men receiving the supplements. A six-month supplementation of 100 µg/d selenium, 30 mg/d vitamin E, and 50 mg/d soy isoflavonoids in men diagnosed with HGPIN has resulted in a stable PSA within three months in 4.2% of the supplemented group, while a decrease in PSA level was recorded in 57.8% [74]. These authors also record that 34%, 48% and 18.3% of these men having either prostate cancer or remaining with HGPIN or no HGPIN or carcinoma respectively after six months. They further recorded that the prostate cancer risk remained at 25% among men having a stable or decreasing PSA due to supplementation (68%) while the risk reaching 52% in the group with increasing PSA after supplementation (32%). Richie et al., 2014 supplemented 200 µg/d selenium as selenomethionine or 200 or 285 µg/d selenized yeast or placebo for nine months in a group of 69 healthy men, in the age range of 20–79 y who were non-smokers and carrying PSA levels (<4 ng/mL) at baseline [75]. The authors report no change in PSA level compared to the baseline in any of the groups consisting of 15–20 men. Contrary to 110 ng/mL baseline median serum selenium (or 113 ng/mL mean serum selenium) level in our cohort, their cohort recorded a higher mean baseline plasma selenium level ranging from 129 to 139 ng/mL. Our cohort gained a post-supplementation median serum selenium levels ranging from 157 ng/mL among the *KLK*rs17632542 CT genotype carriers to 181 ng/mL in *SEL15* rs5845 TT genotype carriers within six months of supplementation, while the Richie et al., (2014) study recorded mean post-supplementation levels above 200 ng/mL within six months for all three types of supplements. Waters and Chiang (2018) discuss the influence of both inadequate as well as excess selenium in prostate health in dogs [76]. The inverse correlation of serum PSA level with increasing serum selenium level in our cohort with *GPX* rs1050450 T alleles could be due to our men with lower baseline serum selenium levels showing a differential benefit compared to men from Ritchie et al. study [75]. We have previously recorded that tobacco smoking is associated with increased levels of serum PSA levels in New Zealand men [77]. The impact of tobacco smoking on prostate tissue blood perfusion and vascular injury have been reported in men with benign prostatic hyperplasia [78]. It is a possibility that increased serum PSA due to tobacco smoking related prostatic glandular epithelium damage could not be circumvented by the selenium supplementation protocol followed by us.

The post-supplementation selenium level showed a significant difference among those carrying the *SEL*15 rs5845 genotypes, with the TT genotype showing the highest level. However, this is contrary to the findings in studies with functional gene constructs, where selenium supplementation response was shown superior with the *SEL15* rs5845 C allele [27]. A US study by Ekoue et al., 2018 have reported that AAs have a higher frequency (34%) of the TT genotype compared to EAs (3.4%) [60], and this genotype distribution in our New Zealand cohort is similar (3.2%) to the latter group. These authors also record that AAs record a lower level of SEL15 protein in their prostate tissue samples and in the serum compared to the European Americans, and that SEL15 levels are generally lower in prostate cancer tissue compared to adjacent benign tissues. They also record that each ng/mL increase in serum selenium level increases the odds ratio for Gleason sum by 2% in men carrying the *SEL15* rs5845 TT genotype, while in men carrying the CC genotype a five-fold decrease in the odds ratio of higher Gleason sum is recorded. Considering the highest retention of serum selenium by supplementation among men carrying the *SEL15* rs5845 TT genotype in our cohort without an accompanying decline in PSA level, they may be a group that should consider serum selenium adjustments cautiously.

The interaction of a panel of dietary nutrients associated with DNA methylation, shows that intakes above the RDI for zinc, benefitting with a significant reduction in PSA levels with increasing selenium levels due to supplementation. Reduction in PSA with increasing serum selenium levels was more stringent with vitamin B12 levels below the RDI. If the selenium supplementation-related PSA reduction was at least partially a consequence of increasing the activity of the antioxidant seleno enzyme *GPX*, it is imperative that nutrient levels of both selenium and zinc be sufficient. The redox action of seleno enzyme *GPX* is preceded by prior production of H_2_O_2_ through the superoxide dismutases. Such superoxide dismutases include the mitochondrial MnSOD and cytosolic Cu/Zn SOD, which catalyze reduction of the superoxide radicals to H_2_O_2_. Therefore, besides being an important molecule in the DNA methylation process [41], zinc is also an important trace mineral in the Cu/Zn SOD metaloenzyme, supporting redox cycling of reactive oxygen species. Findings of the 2008/09 New Zealand Adult Nutrition Survey indicates that 39% of men had inadequate intakes of zinc (median daily intake 12.9 mg) while in the category of men >71 years, this inadequacy was common among 89.7% of men [79]. The New Zealand Adult Nutrition Survey indicates that 32% of the New Zealand males also have inadequate intakes of selenium (67 μg/d that is closer to our current estimate of 65 μg/d). The comparative dietary intake of selenium recorded for US males >20 years was 135.9 μg/d [80]. The major source of selenium in New Zealand diets comes from breads (15%) followed by fish, and sea food (12%) and poultry (10%) [79]. An inadequate intake of vitamin B12 recorded in the 2008/09 New Zealand Adult Nutrition Survey [79] was only 1.3% for men whereas our cohort had 4.4% men with this inadequacy. The above details indicate the importance of dietary nutrient adjustments in achieving optimum prostate glandular epithelial health benefits of selenium as shown through the surrogate serum PSA marker.

Our study implies that it could be beneficial for New Zealand men carrying the *GPX1* rs1050450 T alleles, to get their serum selenium levels adjusted, at least to reach the level between 116 and 150 ng/mL as estimated by us as the optimum level for DNA integrity [15,45]. Besides, our data indicate the importance of serum selenium level adjustments in men carrying the *GPX1* rs1050450 T alleles for prostate glandular epithelial architecture stability that can be monitored with serum PSA levels, rather than using a set dose of selenium supplements. This study also highlights the importance of other nutritional requirements such as dietary zinc and vitamin B12 in maximizing selenium related benefits on prostate health.

Among the shortcomings of the current study are the duration of the study being limited to six months, the nutrient measures except serum selenium were assessed based on the four-day diet and activity diaries that may not provide an accurate nutrient profile in blood of the participants. Another shortcoming is the nutrient profiles were analysed only with a limited panel associated with DNA methylation. It will be beneficial to assess a wider range of serum/plasma based nutrient profiles for better understanding of the interactive nature of nutrients with selenium supplementation and PSA outcomes. Our study has also not recorded urinary selenium excretion, which would have further supported our claim on variable supplemented selenium retention levels based on baseline serum selenium level. As prostate cancer has a long aetiology, it is important that men from this New Zealand selenium supplementation study be prospectively monitored for any future prostate cancer recordings, and to understand the interactive genetic, nutrient and serum PSA measures in such prostate cancer outcomes.

## 5. Conclusions

The current analysis shows the interactive influence of supplemented selenium with demographic, lifestyle, genetic and dietary factors, on prostate glandular architecture stability measured through serum PSA. This highlights the importance of optimizing serum selenium levels on a personalize scale, rather than depending on a continuous single dose selenium supplement for prostate health benefits. However, as we do not have access to participant-reported parameters that allows determining whether it influenced “prostate health” (such as decrease episodes of prostatitis and voiding issues among urology subgroups as well as effects on semen), conclusions based only on PSA levels from this study should be considered carefully.

## Figures and Tables

**Figure 1 nutrients-12-00002-f001:**
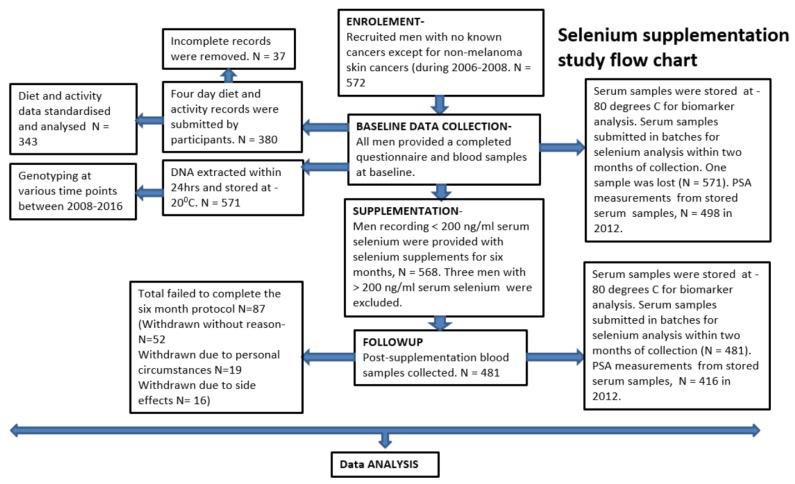
Flow chart of the study.

**Figure 2 nutrients-12-00002-f002:**
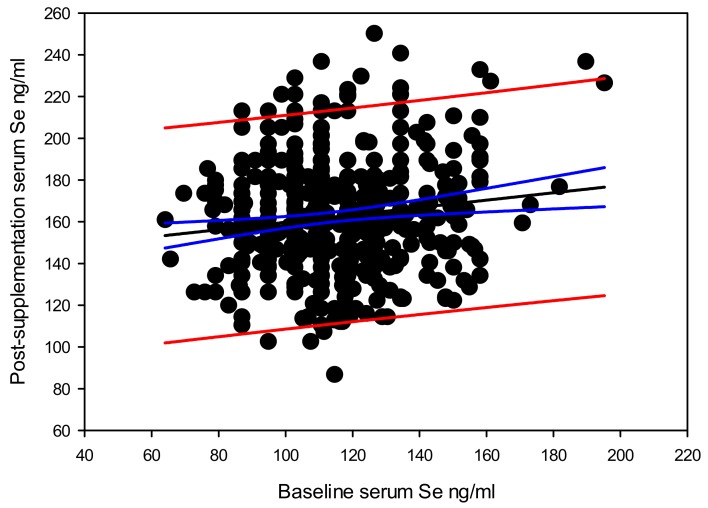
Variation between baseline and post-supplementation serum selenium levels (regression line shown in black, 95% confidence intervals shown in blue and 95% prediction bands shown in red).

**Figure 3 nutrients-12-00002-f003:**
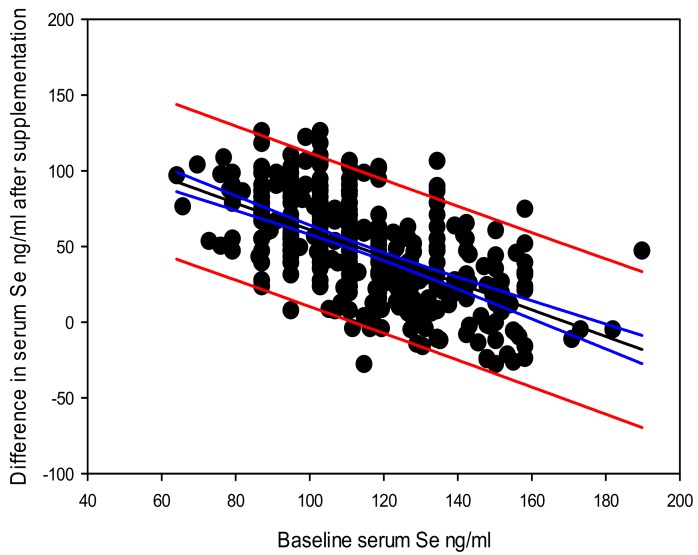
Variation in difference in the serum selenium levels after selenium supplementation and the baseline serum selenium levels (regression line shown in black, 95% confidence intervals shown in blue and 95% prediction bands shown in red).

**Table 1 nutrients-12-00002-t001:** Participant summary characteristics at baseline for all and those who completed the supplementation protocol.

Character		Baseline All	Protocol Completed	*p* Value
Continuous variables				
		Median (25th and 75th percentile) (number)	Median (25th and 75th percentile) (number)	
Age (years)		54 (44.0, 63.8) (572)	55 (46.0, 65.0) (481)	0.11
BMI (kg/m^2^)		26 (24, 29) (548)	26 (24, 29) (463)	0.93
Baseline serum Selenium level ng/mL		110.5 (94.7, 126.3) (571)	110.5 (98.7, 126.3) (481)	0.62
Baseline serum PSA level ng/mL		0.9 (0.6, 1.9) (498)	1 (0.6, 2.0) (416)	0.70
Categorical variables				
Tobacco smoking (number and %)	Ever	190 (33.2)	166 (34.5)	1.00
	Never	382 (66.8)	315 (65.5)	
Alcohol consumption (number and %)	Yes	492 (86.0)	416 (86.5)	1.00
	No	80 (14.0)	65 (13.5)	
Health disorders (number and %)	None	348 (60.8)	286 (59.5)	0.23
	Cardiovascular	119 (20.8)	104 (21.6)	
	Diabetes	6 (1.0)	6 (1.2)	
	Depression/Anxiety	18 (3.1)	15 (3.1)	
	Inflammatory	26 (4.5)	20 (4.2)	
	Urology	30 (5.2)	29 (6.0)	
	Other	25 (4.4)	21 (4.4)	
Food and activity diary submission (number and %)	Accepted submissions	343 (60)	320 (66.5)	0.20
	Rejected submissions	37 (6.5)	28 (5.8)	
	Not submitted	192 (33.6)	133 (27.7)	

**Table 2 nutrients-12-00002-t002:** Genotype and allele frequencies of the study cohort.

SNP	Genotype Number (% Frequency)			Allele 1% Frequency	Allele 2% Frequency	HW Statistics (*p* Value)
	11	12	22			
*GPX1* rs1050450 C>T	278 (49.1)	247 (43.6)	41 (7.2)	70.9	29.1	1.93 (*p* > 0.05)
*SEL15* rs5845 C>T	353 (61.9)	199 (34.9)	18 (3.2)	79.4	20.6	2.54 (*p* > 0.05)
*MnSOD* rs4880 T>C	149 (26.3)	274 (48.3)	144 (25.4)	50.4	49.6	0.63 (*p* > 0.05)
*AKR1C3* rs12529 C>G	212 (40.5)	235 (44.8)	77 (14.7)	62.9	37.1	0.81 (*p* > 0.05)
*KLK3* rs17632542 T>C	464 (89.2)	53 (10.2)	3 (0.6)	94.3	5.7	1.18 (*p* > 0.05)

1 Major allele, Allele 2 Variant allele, HW; Hardy–Weinberg equilibrium.

**Table 3 nutrients-12-00002-t003:** Linear regression correlation results between baseline serum selenium (X) and post-supplementation serum selenium level (Y), stratified by single-nucleotide polymorphism (SNP) genotypes.

Genotype		11	12	22
*GPX1* rs1050450 C>T	*r* ^2^	0.023	0.014	0.137
	slope	0.186	0.151	0.516
	*p*	0.019 *	0.095	0.031 *
	*n*	240	202	34
*SEL15* rs5845 C>T	*r* ^2^	0.016	0.030	0.007
	slope	0.149	0.228	0.126
	*p*	0.031 *	0.026*	0.761
	*n*	300	164	15
*MnSOD* rs4880 T>C	*r* ^2^	0.030	0.027	0.008
	slope	0.226	0.207	0.105
	*p*	0.060	0.011 *	0.321
	*n*	121	236	120
*AKR1C3* rs12529 C>G	*r* ^2^	0.018	0.080	0.000
	slope	0.167	0.356	−0.007
	*p*	0.075	<0.0001 *	0.965
	*n*	180	196	64
*KLK3* rs17632542 T>C	*r* ^2^	0.019	0.028	
	slope	0.176	0.193	
	*p*	0.006 *	0.288	
	*n*	391	42	

1 Major allele, 2 Variant allele, *r*^2^ = coefficient of determination, *p* = statistical significance of slope (regression coefficient), *n* = number of individuals tested; * indicates statistically significant.

**Table 4 nutrients-12-00002-t004:** Genetic variation in baseline and post-supplementation serum selenium and PSA levels, given as median (25th and 75th percentiles).

SNP	Test Character	Genotype			*p* Value
		11	12	22	
*GPX1* rs1050450 C>T	Baseline Selenium	110.5 (102.6, 126.3)	110.5 (94.8, 126.3)	110.2 (94.8, 124.4)	0.52
	Post-supplementation selenium	157.9 (143.9, 177.7)	157.9 (146.1, 178.8)	161.5 (142.1, 182.6)	0.9
	Baseline PSA	1.0 (0.6, 2.2)	0.8 (0.5, 1.7)	0.8 (0.6, 1.6)	0.09
	Post-supplementation PSA	1.1 (0.6, 2.3)	0.9 (0.5, 1.95)	0.8 (0.6, 1.5)	0.19
*SEL15* rs5845 C>T	Baseline Selenium	110.5 (97.5, 126.3)	110.5 (94.7, 126.3)	106.6 (100.7, 120.0)	0.92
	Post-supplementation selenium	157.9 (142.1, 174.3)	165.4 (146.3, 181.0)	180.8 (150.0, 201.3)	0.03 *
	Baseline PSA	0.90 (0.5, 1.6)	0.95 (0.6, 2.3)	1.8 (0.5, 3.2)	0.06
	Post-supplementation PSA	0.9 (0.6, 2.0)	1.1 (0.6, 2.1)	1.6 (0.4, 2.7)	0.6
*MnSOD* rs4880 T>C	Baseline Selenium	110.5(94.7, 123.2)	110.5 (98.7, 126.3)	110.5 (94.8, 126.3)	0.44
	Post-supplementation selenium	157.9 (144.1, 173.7)	157.9 (142.1, 181.2)	161.9 (147.7, 177.7)	0.37
	Baseline PSA	0.9 (0.6, 2.1)	0.8 (0.6, 1.6)	1.1 (0.6, 2.28)	0.25
	Post-supplementation PSA	1.0 (0.5, 2.3)	0.9 (0.6, 1.8)	1.1 (0.6, 2.4)	
*AKR1C3* rs12529 C>G	Baseline Selenium	110.5 (94.7, 123.0)	110.5 (102.6, 127.3)	110.5 (102.6, 127.5)	0.13
	Post-supplementation selenium	161.9 (147.3, 177.7)	157.9 (143.1, 178.3)	157.5 (138.8, 181.4)	0.61
	Baseline PSA	0.9 (0.5, 1.7)	0.9 (0.6, 1.8)	0.9 (0.5, 1.6)	0.77
	Post-supplementation PSA	0.8 (0.5, 1.8)	1.0 (0.6, 2.0)	1.25 (0.6, 2.1)	0.22
*KLK3* rs17632542 T>C	Baseline Selenium	110.5 (98.7, 126.3)	110.5 (94.8, 128.3)		0.58
	Post-supplementation selenium	158.7 (146.1, 178.5)	157.13 (141.9, 178.6)		0.48
	Baseline PSA	0.9 (0.6, 1.8)	0.6 (0.5, 1.0)		<0.01 *
	Post-supplementation PSA	1.0 (0.6, 1.9)	0.6 (0.5, 1.6)		0.07

1 Major allele, 2 Variant allele; * indicates statistically significant.

**Table 5 nutrients-12-00002-t005:** Linear regression statistics between serum selenium gain by supplementation (X) and subsequent difference in serum PSA (Y), stratified by participant baseline characteristics.

**Interacted Baseline Character**		**Continuous Variables**	
		**<Cut-off**	**≥Cut-off**
Age	*r* ^2^	0.019	0.002
	slope	−0.002	−0.006
	*p*	0.048 *	0.551
	*n*	202	181
BMI	*r* ^2^	0.013	0.002
	slope	−0.005	−0.007
	*p*	0.110	0.527
	*n*	193	174
Serum selenium ng/mL	*r* ^2^	0.000	0.000
	slope	0.001	0.000
	*p*	0.956	0.976
	*n*	159	224
Serum PSA level ng/mL	*r* ^2^	0.002	0.002
	slope	−0.002	−0.005
	*p*	0.247	0.599
	*n*	205	178
		**Categorical variables**	
		**Ever**	**Never**
Tobacco smoking	*r* ^2^	0.003	0.014
	slope	−0.009	−0.003
	*p*	0.532	0.031 *
	*n*	126	257
Alcohol consumption	*r* ^2^	0.001	0.002
	slope	−0.003	−0.010
	*p*	0.550	0.304
	*n*	334	50

*r*^2^ = coefficient of determination *p* = statistical significance of slope (regression coefficient) *n* = number of individuals tested; * indicates statistically significant. Cut-off for continuous variables were based on the median values at baseline.

**Table 6 nutrients-12-00002-t006:** Linear regression statistics between selenium gain (X) and subsequent difference in serum PSA (Y) after selenium supplementation, stratified by SNP genotypes.

Genotype		11	12	22
*GPX1* rs1050450 C>T	*r* ^2^	0.000	0.031	0.223
	slope	−0.002	−0.006	−0.017
	*p*	0.810	0.022 *	0.011 *
	*n*	186	168	28
*SEL15* rs5845 C>T	*r* ^2^	0.004	0.014	0.000
	slope	−0.002	−0.003	0.010
	*p*	0.340	0.184	0.946
	*n*	241	130	12
*MnSOD* rs4880 T>C	*r* ^2^	0.001	0.018	0.031
	slope	−0.005	−0.002	−0.0098
	*p*	0.799	0.069	0.090
	*n*	102	186	94
*AKR1C3* rs12529 C>G	*r* ^2^	0.000	0.023	0.000
	slope	−0.000	−0.006	0.000
	*p*	0.858	0.063	0.964
	*n*	141	154	53
*KLK3* rs17632542 T>C	*r* ^2^	0.009	0.111	
	slope	−0.003	−0.005	
	*p*	0.092	0.050	
	*n*	309	35	

1 Major allele, 2 Variant allele, *r*^2^ = coefficient of determination *p* = statistical significance of slope (regression coefficient) *n* = number of individuals tested; * indicates statistically significant.

**Table 7 nutrients-12-00002-t007:** Interactive correlation between selenium gain by supplementation and subsequent change in serum PSA levels when stratified by dietary nutrient intake below or above the cut-off levels among healthy men.

Interacted Nutrient		<Cut-Off	≥Cut-Off
Selenium	*r* ^2^	0.010	0.001
	slope	−0.003	−0.001
	*p*	0.200	0.778
	*n*	166	128
Zinc	*r* ^2^	0.000	0.046
	slope	−0.001	−0.005
	*p*	0.801	0.015 *
	*n*	164	130
Vitamin B12	*r* ^2^	0.795	0.003
	slope	−0.020	−0.002
	*p*	<0.001 *	0.385
	*n*	9	284
Folate	*r* ^2^	0.005	0.004
	slope	−0.002	−0.002
	*p*	0.359	0.488
	*n*	167	127
% energy from protein	*r* ^2^	0.002	0.014
	slope	−0.002	−0.003
	*p*	0.610	0.158
	*n*	146	148

*r*^2^ = coefficient of determination *p* = statistical significance of slope (regression coefficient) *n* = number of individuals tested; * indicates statistically significant. Cut-offs were based on the RDI of selenium, zinc, vitamin B12, and dietary folate equivalents, and the median of the % energy derived from dietary proteins.

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
