# Peer review of "Selenium Supplementation and Prostate Health in a New Zealand Cohort"

_nutrients, 2019, doi:10.3390/nu12010002_

Round 1

Reviewer 1 Report

Briefly, the authors analyzed the impact of selenium supplementation on PSA levels in a cohort of men. They used PSA level as surrogate for prostate health and associate this also with the aim to decrease the risk of prostate cancer. Selenium supplementation did not alter PSA levels in general but found a decrease in PSA in younger men, never-smokers, men with high zinc and low B12 intake, as well as a certain genotype.

This is an interesting study and well-written manuscript. Some shortcomings should be addressed, however:

1) Although it is valid to relate PSA as indicator for prostate cancer, the authors should de-emphasize the impact of their findings in this regard. They did not analyze the cohort for prostate cancer and a mere PSA level at baseline is not an indicator for it especially at the low levels detected.

2) The supplementation of selenium was only for 6 months and there is no patient-reported parameter that allows to determine whether it influenced 'prostate health' (such as decrease episodes of prostatitis, voiding issues, influence on semen analyses etc.)

3) As the authors mentioned, the subanalyses regarding dietary co-factors depend on self-reported diaries which leaves room for error yet it is understood that this is due tot he study design.

Author Response

We have expanded the limitations (indicated in bold) as kindly suggested by both the reviewers 1 and 2 as follows.

Among the shortcomings of the current study are, the duration of the study being limited to six months; and the nutrient measures except serum selenium were assessed based on the four-day diet and activity diaries that may not provide an accurate nutrient profile in blood of the participants. Another shortcoming is the nutrient profiles were analysed only with a limited panel associated with DNA methylation. It will be beneficial to assess a wider range of serum/plasma based nutrient profiles for better understanding of the interactive nature of nutrients with selenium supplementation and PSA outcomes. Our study has also not recorded urinary selenium excretion, which would have further supported our claim on variable supplemented selenium retention levels based on baseline serum selenium level. As prostate cancer has a long aetiology, it is important that men from this New Zealand selenium supplementation study be prospectively monitored for any future prostate cancer recordings, and to understand the interactive genetic, nutrient and serum PSA measures in such prostate cancer outcomes.

5. Conclusions

The current analysis shows the interactive influence of supplemented selenium with demographic, lifestyle, genetic and dietary factors, on prostate glandular architecture stability measured through serum PSA. This highlights the importance of optimizing serum selenium levels on a personalize scale, rather than depending on a continuous single dose selenium supplement for prostate health benefits. However, as we do not have access to participant-reported parameters that allows determining whether it influenced 'prostate health' (such as decrease episodes of prostatitis and voiding issues among urology subgroups as well as effects on semen); conclusions based only on PSA levels from this study should be considered carefully.

Reviewer 2 Report

The authors have evaluated the selenium supplementation in prostate health, demonstrating that the benefits of selenium is dependent of lifestyle and genetic factors. Overall, this article highlights the importance of these factors in order to optimize the selenium supplementation. However, there are some limitations that should be inserted in conclusions section. In addition, it should be added in introduction more information about SNPs (for example, why were selected, the location inside the gene, if affects the protein or not).

I have assumed the description of figure 2 and 3 as correct, because these graphs are missing on article.

Author Response

As kindly suggested by the Reviewer 1, we have added the following paragraph giving more details of the SNPs.

The glutathione peroxidase 1 (GPX1) is a selenoprotein encoding gene located in the chromosomal region 3p21.3 and records rs1050450 C>T SNP that is responsible for a nonsynonymous amino acid change from proline to leucine [21, 22]. The catalytic activity of GPx is affected by the rs1050450 T allele [23], while this allele is also associated with several cancers [13, 24, 25]. The selenoprotein 15 (SEL15) encoding gene is located in chromosomal region 1p31 and records rs5845 C>T SNP [26, 27]. This rs5845 C>T SNP which relates to the amino acid position 811 is located in a selenocysteine insertion sequence (SECIS)-like structure within the SEL15 protein, and is associated with the rs5859 G>A SNP which relates to the amino acid position 1125 located within the SECIS element [27]. According to these authors, the latter polymorphism influences selenocysteine incorporation in SEL15. The SEL15 rs5845 and its associated SNP rs5859 are associated with several cancers [13, 28, 29]. The mitochondrial manganese superoxide dismutase (MnSOD) encoding gene is located in the chromosomal region 6q25.3 and the rs4880 C>T SNP produces a nonsynonymous alanine to valine amino acid change [30, 31]. This rs4880 polymorphism also has been associated with various cancers [32, 33]. The aldo-keto reductase 1C3 (AKR1C3) encoding gene is located in the chromosomal region 10p15 and the rs12529 C>G polymorphism produces a nonsynonymous amino acid change from histidine to glutamine [34]. This rs12529 polymorphism is also associated with various cancers [20, 35, 36]. The kallikrein-related peptidase 3 (KLK3) gene encodes the serine protease PSA and this gene is located in the chromosomal region 19q13 [37]. The KLK3 rs17632542 T>C is a nonsynonymous polymorphism with an amino acid change from isoleucine to threonine and has an association with prostate cancer risk as well as with the PSA level [20, 38].

Round 2

Reviewer 1 Report

Overall, the comments have been reasonably addressed.